# The Daily Mile: teachers' perspectives of the barriers and facilitators to the delivery of a school-based physical activity intervention

Stephen Malden,[1,2] Lawrence Doi[1]

[1]Scottish Collaboration for Public Health Research and Policy, School of Health in Social Science, University of Edinburgh, Edinburgh, UK
[2]Physical Activity for Health Group, University of Strathclyde School of Psychological Sciences and Health, Glasgow, UK

**Correspondence to**
Stephen Malden;
stephen.malden@strath.ac.uk

## ABSTRACT

**Objectives** Children spend a significant amount of their time in a school environment, often engaged in sedentary activities. The Daily Mile is a physical activity intervention which aims to increase physical activity and fitness in children through the completion of an outdoor teacher-led walk or run during the school day. This study aimed to explore the barriers, facilitators and perceived benefits of the Daily Mile from the perspectives of teachers through the use of qualitative semi-structured interviews. It also aimed to identify important context-specific factors, which might require consideration for those who intend to adopt the Daily Mile.

**Setting** Eight Local Authority primary schools in the City of Edinburgh and East Lothian, UK.

**Participants** Thirteen teachers (eleven women) who teach children in primaries one to seven in a school which delivered the Daily Mile.

**Results** Data were analysed using an interpretative thematic analysis. Teachers were positive and enthusiastic about the Daily Mile and perceived it to be beneficial to children's health and fitness. A number of barriers to participation were identified including inadequate all-weather running surfaces and time constraints in an already full school curriculum. The perceived impact on learning time was identified as a concern for teachers, while other benefits were also identified including increased teacher–child rapport and perceived enhanced classroom concentration levels.

**Conclusion** The Daily Mile appears to be a valuable addition to the school day, however important context-specific barriers to delivery of the Daily Mile exist, which should be considered when implementing the Daily Mile in schools.

## INTRODUCTION

A lack of physical activity is a determinant of numerous health conditions in childhood.[1] Specifically, sedentary behaviours and low levels of physical activity are partly to blame for the marked increases in childhood obesity that have been observed internationally in recent years.[2 3] In the UK, childhood overweight and obesity levels are high, approximately 30%.[4] Additionally, physical

## Strengths and limitations of this study

► One of the first studies to investigate teachers' views of the Daily Mile.
► Sampled teachers from schools in multiple locations to investigate contextual factors.
► Data collection with children would have allowed for data triangulation.
► Small sample size may limit generalisability of findings.

activity levels are low[5] and have been shown to decrease with age, a trend that has been similarly observed in other European countries and the USA.[6] Considering this, there is a need for the implementation of effective, evidence-based interventions to increase physical activity levels in school-aged children.

The Daily Mile is an example of a school-based intervention that involves teachers taking their class outside at some point during the school day to participate in 15 min of exercise. Children choose what pace they complete the activity, with those who run for the full 15 min likely to have completed approximately one mile (approximately 1.6 km). The intervention is intended to be delivered in addition to the 2 hours per week of quality physical education as required by the Scottish Curriculum,[7] and is not intended to replace this time. The Daily Mile was originally introduced to primary schools (which teach ages ~5 to ~11 years) to increase fitness levels,[8] but it may also have a positive effect on physical activity, sedentary behaviour and body composition.[9] Since its inception at St Ninian's Primary School in Stirling in 2012, participation and expansion of the initiative has grown exponentially, with the Daily Mile now taking place in more primary schools across Scotland, in addition to a number of sites both within the UK and internationally.[8]

A recent quasi-experimental pilot study by Chesham *et al*[9] showed that when compared with control schools, children who participated in the Daily Mile showed significant improvements in objectively measured daily moderate-vigorous physical activity (MVPA), objectively measured sedentary time, fitness and body composition.[9] To date, the aforementioned study is the only research currently offering empirical evidence of the Daily Mile's efficacy, although another randomised trial is being planned.[10] However, interventions do not usually function the same everywhere, and it is important to understand contextual factors that may influence how they work.[11] Contextual issues have been found useful in other school programmes, where context-specific differences have been considered when testing and developing an intervention.[12] Furthermore, a systematic review of studies which examined factors which influenced the implementation of physical activity policies in schools, found that factors relating to environmental context/resources such as provision of adequate equipment/facilities, and social influences such as teacher/staff support were important predictors of implementation success or failure.[13] Therefore, examining the experiences and perspectives of teachers in early adopter schools would allow policy-makers and practitioners to adapt the Daily Mile to suit their settings where necessary and identify facilitators and barriers to implementation.

A recent qualitative study by Ryde *et al*[14] was the first to assess the context specific factors, which influence the success of the Daily Mile's implementation in four schools in central Scotland. The study concluded that having a supportive organisational environment and allowing teachers autonomy to deliver the initiative facilitated the implementation of the intervention. While the study offers important insight into factors influencing the implementation of the Daily Mile, the authors highlight that it may be biased due to the included teachers being advocates of the Daily Mile and the selection of schools in one geographical area.[14] The authors recommended for future studies to interview more teachers across different

geographical locations in schools which had implemented the Daily Mile with varying degrees of success.[14] Therefore, the present study explored the barriers, facilitators and perceived benefits of the Daily Mile from the perspectives of teachers who deliver the intervention in multiple schools across two geographical locations in Scotland. It also aimed to identify important context-specific factors, which might require consideration for those who intend to adopt the Daily Mile in the future.

## METHODS
### Patient and public involvement
A senior member of teaching staff (physical education coordinator) who had well established contacts with teachers in multiple schools acted as a liaison during recruitment and liaised with potential participants. The research findings will be shared with the participants in a short summary report which will be emailed to them directly and to head-teachers of the participating schools.

### Design
Adopting interpretive approach, qualitative methods were used to explore teachers' perspectives of the Daily Mile and their experiences of delivering it. Semi-structured interviews were conducted, consisting of open-ended questions which aimed to determine how teachers define the Daily Mile, how they deliver the initiative, perceived benefits for the children who participate, and potential context-specific barriers and facilitators to participation. Our goal was to explore their experiences of the Daily Mile in order to inform future practice rather than to develop new theory.

### Setting, participants and recruitment
This study was conducted in Local Authority primary schools within two council areas: the City of Edinburgh and East Lothian, Scotland. The sites contain significant rural and urban populations, while also having both some of the most affluent areas in the country and some of the

**Table 1** Characteristics of schools

| School identifier | Accessible greenspace in school grounds | Accessible all-weather surface in school grounds (such as Astroturf) | Rurality (rural, semirural, urban) | Level of deprivation (SIMD quintile, 1=most deprived 5=least deprived) |
|---|---|---|---|---|
| A | Yes | No | Semirural | 3 |
| B | Yes | Yes | Semirural | 5 |
| C | Yes | Yes | Urban | 3 |
| D | Yes | No | Rural | 4 |
| E | Yes | No | Urban | 1 |
| F | No | Yes | Urban | 1 |
| G | Yes | No | Rural | 5 |
| H | Yes | Yes | Semirural | 1 |

SIMD, Scottish Index of Multiple Deprivation.

most deprived areas, as defined by the Scottish Index of Multiple Deprivation in table 1.[15]

Participants were primary school teachers who taught years 1–7 at schools which either currently, or formerly, participated in the Daily Mile. As it is mostly class teachers who deliver the Daily Mile, head-teachers or teaching staff who do not have regular prolonged contact with children were excluded from participating. A total of thirteen teachers took part in the study, consisting of eleven women and two men.

A purposive sampling strategy was used to recruit teachers. Specifically, contact was made with a physical education coordinator in one of the Local Authority areas who acted as a liaison between the research team and schools/teachers. Between August 2016 and February 2017, the liaison delivered study information sheets and personally spoke to selected head-teachers about the study. Schools/head teachers were approached based on the liaison's knowledge that they were a school that had been delivering the Daily Mile. In order to enhance participants' recruitment, emails and letters were sent to head teachers of additional primary schools. This contained study information sheets and researcher's contact details. Twenty-one schools were approached to participate in the study, for which ten initially agreed to participate. Teachers from eight of the ten schools subsequently agreed to participate in the study.

## Data collection

Semi-structured interviews were conducted face-to-face in a private room within the participating schools by one researcher. Both researchers have extensive experience in conducting qualitative interviews with a broad range of participants for public health evaluation purposes, and have undertaken formal training in qualitative data collection. Participants were asked to give a brief history of the Daily Mile at their school, before probing questions were asked regarding subjects such as how the initiative is viewed by the staff at the school, approaches taken to deliver the programme and any perceived benefits, facilitators or barriers associated with participation and delivery. A sample of the interview topic guide can be viewed in online supplementary appendix 1. Interviews were usually conducted after the school day and lasted between 21 and 58 min. All interviews were audio-recorded via digital audio recorder and transcribed verbatim.

## Data analysis

Using interpretive approach, this study employed thematic analysis[16] to analyse the qualitative data that were generated following verbatim data transcription. After familiarising ourselves with the transcribed data, a coding framework was developed by both authors who independently coded the same two most informative transcripts from the dataset, before collaborating to refine the coding framework. Both authors independently read and re-read each remaining transcript and used the coding framework to assign codes to the text. New codes identified were discussed and added to the framework. This ensured consistency and rigour. Subsequently, similar codes were grouped together into overarching themes, which were reviewed and refined to ensure the coded data adequately reflected themes that encompassed them. Interpretation of data was undertaken by comparing similarities and differences within and across themes. Data analysis was facilitated by NVivo V.10 qualitative data analysis software.

## RESULTS

### Characteristics of schools

The characteristics of the eight schools where the 13 teachers were drawn from varied with regard to facilities, rurality and level of deprivation as detailed in table 1. All but one school had access to sufficient greenspace within the school grounds, while half of the schools had access to an all-weather surface such as an Astroturf football pitch or a solid polyurethane running track. Three schools were located within a large town or city, three were in a semirural location such as a small country town and two schools were in a rural location. Three schools were located within areas of high deprivation, three within areas of medium deprivation and two in affluent areas.

### Themes

Eight themes were identified, and are detailed below.

### Style of delivery

The style of delivery varied considerably between teachers and schools. While each school followed a similar Daily Mile protocol for delivery, some schools offered more autonomy to teachers concerning how and when they delivered the Daily Mile.

> So yeah, it does happen in this school and what we do is that we split it, we don't do the mile in one go, we do half a mile in the morning and half a mile in the afternoon and it's immediately after break, or immediately after lunch, with the exception of Thursdays when it really works in our favour because we do it immediately after assembly. Because the kids have been sitting down and just not, being in a stuffy hall for, you know, half an hour. F2

There were also some differences in the frequency and level of engagement with the Daily Mile among the schools. A number of participants stated that all classes at their school took part, often at the request of the head teacher, while others mentioned that only some classes did the Daily Mile and that participation does not happen every day either due to teacher preferences or age/ability of the children.

> Some of the other classes do it every day as well, but some just do it every now and then. F1

> I know that the [primary] ones certainly don't do it because it's a lot more difficult to manage. B2

There were slight variations in how the Daily Mile was defined by the teachers. Variations in what constitutes participation were evident, with some teachers defining the Daily Mile as 15 min of exercise, and others as an exact mile that is done to completion regardless of time taken. The emphasis on running the Daily Mile also differed between teachers and schools, with some teachers' strongly encouraging running, while others were happy for children to choose whether to walk or run.

> I don't start by saying, you can run or walk it, I start by saying, it's a run and I sneakily tell people they can walk if I think that, you know, they need to. But even if you tell them to walk, quite often they end up running. D2

The time at which the Daily Mile was delivered during the school day also differed among the teachers. Some participants stated that they purposefully delivered the intervention at set times during the day in order to establish a routine or get a desired behavioural or performance-based response from the children.

> I'm doing it after break and lunch. I think that's really effective because you don't need to take them out. There's no difference. They're already outside. They can take their jackets off or they can leave them on. F1

> I've done it in the middle of the afternoon rather than straight after break – I much prefer that. I prefer to use it as a break in the lessons or change of lesson. D1

There was variability among the sample regarding the interaction between the Daily Mile, and formal physical education sessions. A number of teachers explicitly stated that they do the Daily Mile in addition to the 2 hours per week of physical education recommended by the school curriculum. However, a few of the teachers asserted that they count the Daily Mile as part of the physical education curriculum and therefore as a part of the 2 hours of weekly PE which they are expected to deliver.

> See, I didn't know that [the Daily Mile does not count towards PE time]. I've sometimes done it as a warm-up, so I've done the Daily Mile and then 45 min of PE and then don't have to do a warm up in PE. D1

> I think people were concerned about where would it fit, in the timetable, because obviously we're stretched for our 2 hours of PE, and now we have to do the Daily Mile. Last year I did it as part of my PE. B2

### Motivating children to participate

Approximately half of the teachers used some form of motivational or incentive/rewards-based system with children when doing the Daily Mile. Such techniques included the handing out of tokens for each lap of the running area completed, or tracking the distance ran over an accumulative time-period and matching it with the distances to various geographical locations.

> With the Primary 1s and 2s and 3s last year, we gave them…you know the Unifix Cubes that stick together? So every time they complete a lap, they get a cube. And they add them on. And it was good in so far as we could use that then as a numeracy lesson and obviously for the little ones, you know, they were counting, they were being inspired. D1

> We have a big board at the front of the school and it shows everybody's progress tracked. We are aiming to go to different places in the world and to the moon as a whole school. It's really positive and they are motivated. H1

Some teachers stated that they found that their own participation in the Daily Mile motivated the children to try to keep up or beat them, thus increasing the children's engagement with the activity.

> Every now and again I'll go and run with them. But they'll always, you always see a boost in them when, 'the teacher's coming, the teacher's coming, he's coming!' And then I'll just go, oh come on guys, I'm beating you and I'm so much older. And it just adds something to it. F2

### Health, well-being and fitness

A number for the teachers attributed the Daily Mile to improvements in general health and fitness that they observe among children in their care. Specifically, the majority of the teachers indicated that the children's time taken to complete the Daily Mile had generally improved since the programme started, and others attributed school success in sporting events to Daily Mile participation

> We interviewed some of the children about how they'd got on [competing in an athletics competition]. Actually their reflections were, we think we did better because we are fitter because we do the mile and that wasn't prompted at all, so I think that they feel fitter and healthier. C1

> They are definitely fitter because I forgot to say that we are recording last year's Primary 5s at the start of the year and we did fitness levels with them before we even started the mile. We've done the same cohort of children every term since and we have that recorded as well for every single child's results we can see that they are all fitter, so that's pretty good. H1

Two teachers also speculated that Daily Mile participation led to improvements in asthma symptoms among some children. One of the teachers explained this below.

> I'm seeing a great difference in kids with asthma. Huge. Loads of them, all the time. I've seen kids that needed inhalers every 2 min and because they're doing this run, they're saying, I feel brilliant, I feel really good. And I've had that from, oh quite a few kids. F1

## Concentration

There were mixed, but slightly positive findings regarding how the Daily Mile affected pupils' concentration levels. Some teachers reported that they had not noticed any obvious changes in concentration levels since the Daily Mile was introduced.

> I definitely couldn't say that, you know, their concentration is better because I do it at weird times of the day. It doesn't help unfortunately. And I would love to do it halfway through the afternoon, but it's just not feasible. And to get them settled and back down again, I don't know if that's worse… F2

However, others stated that they have observed noticeable improvements in concentration and would often choose to do the activity at a specific time of the day when concentration levels are expected to be low.

> Doing it in that halfway between break time and lunchtime, you could see a dip in concentration coming, we would go outside, just ten minutes outside, we have a run around, we come back in and then we'd go back to our job and we were maybe a bit more prepared for a bit more learning and thinking. B1

## Behaviour

The impact of the Daily Mile on behaviour was a contentious issue among the participants. Some teachers argued that the Daily Mile made little or no difference to disruptive behaviour of pupils.

> I really didn't have behaviour issues last year, this year I've got loads of them and I can't honestly say that there is huge difference yet. D1

> I still have the same disruptive behaviours on the days that I do it [the Daily Mile] as on the days that I don't do it, unfortunately. E1

However, two teachers felt that the Daily Mile did improve behaviour in the classroom. They attributed this to the release of excess energy, which often made the children restless at certain times of the day. The example below by one of the teachers explains this point.

> And from a behavioural point of view there was one particular boy who was really switched off, you know, but the running really…he performed really well in that and it, sort of, really made a positive impact on how he felt about himself and how he performed generally in school. C1

## Teacher–pupil rapport

Approximately half of the participants stated that a major benefit of the Daily Mile was that it allowed them the opportunity to build a rapport with the children in their class in a way that is difficult to do in a classroom environment.

> It's the pastoral side of it. Kids will tell you things while you're out there doing that that they won't tell you in a classroom setting, or that they won't disclose to another adult. Not, kind of, generally speaking, you know, kind of big scary disclosures, but they'll just be more open and honest with you. A1

> It's also nice for teachers, it gives you a chance to run alongside and walk alongside some of the children and have a chat about something that may have happened in class, or you just want to have a chat with them, because it's not something we get to do one to one in the class, we don't have time. E2

Teachers acknowledged that the Daily Mile offered a unique opportunity to talk about issues with certain individuals, or to get to know the children in their class better.

> It's a really social time because you get a chance to speak to the kids, or you can deal with any issues. I think actually along with them running, because every lap, a couple of them might stop and chat for a second. F2

## Perceived impact on learning time

In general, the Daily Mile was considered a worthwhile initiative due to the known benefits of physical activity on health and well-being. However, the majority of the participants conceded that it is time-consuming, and reduces the amount of time they have available to dedicate to other subjects in the classroom per week, which may potentially affect learning.

> The biggest challenge is the fact that it takes out from other learning time, and that can be really challenging because most teachers are already feeling like they're trying to cram too much into the day. So to add another thing into the day is really difficult, even though you know it's important, you want them to do maths and read and write and all those other important skills as well. It's just trying to fit everything in. E1

Furthermore, some participants were of the opinion that as the Daily Mile is not supposed to count as PE, they struggle to provide both activities in full to their class. Teachers stated that this was a contentious issue among colleagues when discussing the Daily Mile, and a number of participants mentioned that some colleagues refused to do the Daily Mile or have complained about it due to time constraints in terms of delivering a full curriculum.

> We're so stressed for time. You have to do it every day. If you've got a PE lesson, then you have to have a Daily Mile as well. D1

A number of teachers expressed the view that getting the children ready to go outside was time-consuming, and added additional time constraints in addition to the Daily Mile itself.

> The other issue is, these children here change their shoes and it takes so long to change shoes to go

outside and change back that you're losing a chunk of time. G1

## Weather and resources

All the participants identified the weather as a barrier to participation; specifically wet and icy weather. However, the majority of teachers felt they would like to do the Daily Mile in all weathers, but are limited by unsafe running surfaces, and inappropriate footwear/weather gear worn by some children.

> We often have howling gales here and so we probably wouldn't go out in howling rain. G1

As previously mentioned, some teachers highlighted the running surface as a limitation. Specifically, schools with large greenspace were viewed as suitable for participation. However, this is not necessarily safe to run on in wet conditions, which limited the ability of some schools to do the Daily Mile in wet weather due to not having access to alternative facilities.

> There is a hard tarmac and there is a grass field, I mean, they use the grass field part of the playground as long as it's dry. As soon as it's wet we can't use that, so it's actually on a rough tarmac it's not a good running surface at all. D2

Alternatively, teachers at the school with insufficient greenspace felt that this limited their ability to do the Daily Mile due to insufficient space and inadequate surfaces.

Teachers that had access to artificial playing surfaces or all-weather surface generally believed that their school had better resources for Daily Mile participation than those without.

> We can use it (3G pitch) in all weathers and, you know, they don't need to get changed, they are not having to worry about mud and things like that. C1

While facilities were deemed important to participation, in general the teachers found ways to adapt the initiative to suit the facilities that were available at their school. This sometimes meant not participating in severe weather, or using indoor space such as a gym hall as an alternative.

Another barrier identified was that a number of teachers felt that some children were not properly equipped with the appropriate clothing to take part in the Daily Mile in all weathers.

> The kids don't all have wet weather gear. Loads of kids who don't have…like, I mean, this is the first week where all the children in my class have worn a coat. So I would say that's much more of a barrier than having the space. E2

## DISCUSSION

The Daily Mile offers an innovative approach to physical activity in schools, and similar to another recent study,[14]

this study has identified important factors that influence the implementation of the Daily Mile, and has added further to the knowledge-base regarding the context-specific barriers and facilitators related to the intervention from the perspectives of teachers. We found that teachers were generally positive about the initiative, and felt the intervention was warranted considering the health benefits of physical activity, and its potential to reduce childhood obesity. However, a considerable number of teachers were concerned about the time-consuming nature of the Daily Mile and the impact of this on learning time.

Although the Daily Mile follows a generic protocol and procedure for implementation, schools and teachers have autonomy over when and how they execute it. Our findings show that some teachers did not undertake the Daily Mile every school day. A growing evidence base is demonstrating that a dose-response relationship exists between physical activity and its associated health benefits. Specifically, improvements in health outcomes tend to increase along with the levels of physical activity.[17–19] Interestingly, some of the teachers in our study stated that they or their colleagues would occasionally count the Daily Mile as part of the compulsory 2 hours of quality physical education time, which may offset the benefits of having the Daily Mile as a stand-alone activity-as it is intended.[8] Additionally, such practices may reduce the wider benefits associated with physical education such as motor skill development and the social/cognitive benefits of sport and team games which may not be addressed by Daily Mile participation.[20] Despite this, Chesham et al.[9] found that compared with control schools, the Daily Mile intervention schools increased MVPA by about 9 min per day, indicating that the intervention does increase PA and in turn may have potential health benefits. However, whether similar findings would be observed in schools which participate in the intervention less regularly, or combine the Daily Mile with physical education time may merit further investigation.

Despite the fact that teachers felt motivated to deliver the Daily Mile, the study also showed the Daily Mile increased pressure on teachers' workload and they were concerned about its detrimental effect on learning time; a finding that is supported by studies of other school-based physical activity initiatives.[21 22] This was in contrast to a similar study in Italy, which also examined teachers and students' attitudes to an outdoor active break programme in a middle school.[23] The study reported that teachers found the outdoor active break programme easy to organise and that it did not negatively influence their teaching activities. It is possible that any differences in the attitudes of teachers in the aforementioned study and our study are due to the age of children involved. While we used primary school teachers, with pupils usually aged 5–11 years, the Italian study used middle school teachers, who taught students aged about 12 years. A recent historical account indicates that increasing focus on academic achievement decreases physical activity opportunities in schools.[24] With academic achievement under constant scrutiny in most countries,

it appears that evidence to ascertain whether the Daily Mile promotes cognition and academic achievement could assure primary school teachers that time invested in the Daily Mile would engender additional educational benefits. Current evidence suggests that physical activity can show a positive effect on constructs related to academic achievement.[24–26] However, recent studies have also demonstrated that academic achievement is more strongly associated with sedentary time than physical activity.[27 28] These conflicting findings highlight the fact that at present, little is known regarding the optimum balance of sedentary time and physical activity for both academic performance and health. Future evaluations of school-based PA interventions similar to the Daily Mile should therefore incorporate measures of academic performance into their design.

Another important finding of the study was that Daily Mile offered social time for teachers and children. This was viewed as important in building a more supportive teacher–pupil relationship. A study has suggested that such supportive relationships are characterised by warmth, trust and low degrees of conflict and were associated with positive school outcomes for children with behavioural issues.[29] Positive teacher–pupil relationships can therefore improve student motivation to learn as well as their academic achievement.[30] The effect of the Daily Mile on teacher–pupil relationships is an interesting finding and should be explored further in future studies.

It was clear that the weather had implications in terms of participation for schools without adequate outdoor facilities, which is consistent with the findings of the review by Nathan et al[13] which demonstrated that factors such as inadequate facilities and adverse weather were identified as barriers to implementation of school-based interventions in seven studies. The Daily Mile is an outdoor school-based physical activity programme. Therefore, schools in temperate regions where the weather is often unpredictable during the winter months should consider improvement to their outdoor facilities in order to have the best chance of fully delivering the Daily Mile throughout the school year.

From 1998, the prevalence of overweight, including obesity among children aged 2–15 has fluctuated between 28% and 33%, and was 29% in 2016.[31] This high prevalence of overweight and obesity highlights the fact that existing interventions are not having the anticipated impact, at least at the population level. Therefore, innovative school-based physical activity programmes such as the Daily Mile require attention in order to understand how they can contribute to the reduction of childhood obesity; which is a major target within current UK and international health policy.[32] Such programmes replace sedentary behaviour and light physical activity with MVPA and tend to reduce adiposity.[33] It is in this respect that the Scottish Government is promoting the Daily Mile initiative more widely across schools in Scotland. The concept of the Daily Mile has also generated international interest, yet until recently there was little evidence regarding its efficacy. The Chesham et al study has provided some assurance of the Daily Mile's benefits to children in terms of increasing levels of MVPA, reducing sedentary time, increasing physical fitness and improving body composition. Certainly, it would be interesting to see whether such benefits can be sustained in the longer-term and how they translate into meaningful reductions in childhood obesity at the population level.

## Strengths and limitations

The study used qualitative interviews to generate a deeper insight into the perspectives of teachers who have crucial role in the delivery of Daily Mile in Schools. However, only the accounts of teachers were used to understand these barriers and facilitators. Using additional data from pupils would have allowed triangulation of data, which would have enhanced our findings. We struggled to recruit teachers into the study, so it is possible that teachers who were likely in favour of the Daily Mile participated. This may imply that the level of positivity depicted by the study may not reflect the overall opinion of teachers who deliver the Daily Mile. On the contrary, it could also signal that any barriers identified were important concerns, which require attention as the Daily Mile continues to expand internationally. Future studies should measure the attitudes and practices of teachers to the Daily Mile using quantitative approaches such as surveys, which can reach a larger sample.

## CONCLUSION

Given that existing obesity prevention strategies are struggling to stem the tide of childhood obesity, innovative school-based physical activity interventions such as the Daily Mile present promising opportunities for policy change within education. However, any policy change should be underpinned by evidence if the practice of the Daily Mile is to be adopted more widely. This study has identified important barriers and facilitators from the perspectives of teachers who play a crucial role in the delivery of the Daily Mile. Our findings indicate that teachers were motivated to deliver the Daily Mile and were positive about its potential to improve physical activity and related health outcomes. However, they were also concerned that it was negatively affecting learning time.

**Acknowledgements** We thank Marion Barclay for her assistance with recruitment for this study, and both Dr Josie Booth and Dr Gemma Ryde for their comments on the interview topic guide. We also thank all the participants who took part in our study.

**Contributors** SM and LD designed the study and conducted data collection with participants. Both authors analysed the data, and SM drafted the first draft of the manuscript. SM and LD revised and prepared the final draft of the manuscript for submission.

**Funding** This work was supported by a University of Edinburgh Innovation Initiative Grant under Grant number GR002292. The authors were both employed by the Scottish Collaboration for Public Health Research and Policy (SCPHRP), University of Edinburgh. At the time of the study, SCPHRP's core Grant was from the Medical

Research Council (Grant Number MR/K023209/1) and the Chief Scientist Office of Scotland.

**Competing interests** None declared.

**Patient consent for publication** Not required.

**Ethics approval** Ethical approval was granted by The University of Edinburgh's Usher Research Ethics Group, and Local Authority research in schools permission was obtained prior to recruitment.

**Provenance and peer review** Not commissioned; externally peer reviewed.

**Data sharing statement** Ethical restriction by the University of Edinburgh Centre for Population Health Sciences Research Ethics Committee prohibits the authors from making the minimal data set publicly available because data contain potentially identifiable information. However, data are available from the corresponding author on reasonable request. Data will be held for five years from February 2017 before being permanently destroyed. Contact: StephenMalden-stephen.malden@strath.ac.uk.

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
