## [Reviewer comments · BMJ Open]

ARTICLE DETAILS

TITLE (PROVISIONAL)	The Daily Mile: Teachers' perspectives of the barriers and facilitators to the delivery of a school-based physical activity intervention
AUTHORS	Malden, Stephen; Doi, Lawrence

VERSION 1 – REVIEW

REVIEWER	Carol Maher University of South Australia, Australia
REVIEW RETURNED	14-Nov-2018

GENERAL COMMENTS	Overall comment: This is an interesting paper, and reveals information which will be useful to inform future implementation of the Daily Mile program in school. More detail on interview methodology and analysis would improve rigour. Overall, this is a modest, scientifically sound study. Generally, I am used to qualitative studies such as this being grounded in a theory, and providing more details regarding their analysis methodology. Despite this, the paper appears to have generated logical, balanced perspectives which will help progress research translation. Comments: 1. Abstract – Structure of the abstract is a little unusual (there is no “methods” section – perhaps this is a journal requirement). At present, methodological detail is dispersed through other sections of the abstract (e.g. the study design is described in the objectives section, and the analytic approach is described in the results). Recommend moving these to create a methods section.2. P3 line 4 - typo - teachers need apostrophe “teachers’3. St Ninian's primary school - capitalise primary school (LINE 24-5)4. p 6 line 9-10 Primary school teachers who teach primaries one through seven at a school, which participated in the Daily Mile or had done so in the past were recruited from the participating schools. This sentence is rather awkwardly worded.5. Perhaps change it to Participants were primary school teachers how taught years 1 through to 7 at schools which either currently, or formerly, participated in the Daily Mile.6. Line 16 and other places – a teacher acted as a "gatekeeper" to others – this word is not appropriate for scientific writing. Suggest replacing throughout the manuscript with either liaison or facilitator.7. P 17 - "personally spoke to selected head-teachers" – on what basis were such head teachers selected? The methods should be reported in a level of detail that is reproducible.
--

	8. P 7 – Data collection: How were the interviews recorded? What training did the interviewer have? Some more information about the interview procedure would be helpful. 9. P7 – data analysis: It appears that coding was determined based on the first two transcripts, and then applied to all remaining transcripts. I have never heard of this analysis approach. Please add citations to support this approach. How would this approach handle new topics that arise in subsequent transcripts that were not present in the first two transcripts? 10. p 7 lines 16-17 “The analysis proceeded to the interpretive phase where similarities and differences between units of the data underwent dynamic series of negotiations before patterns of meaning and understanding developed” – this sentence is very jargony. What does this actually mean? 11. P 6, line 13 - 11 women, 2 men. Abstract says 10 women. 12. P17, line 12 - schools should be lower case 13. Discussion: paragraph 2 of the discussion is essentially the "implications" paragraph, and would make more sense if it was placed nearer the end of the discussion. 14. Paragraph 3 makes an interesting point about the Daily Mile apparently substituting PE for some teachers. While the Chesham study found that the Daily Mile program increased MVPA overall, it is possible that the program was implemented in a more "pure" form in the RCT schools, and that in its more normal translation, replacement of PE is actually occurring. This is an important, and potentially detrimental, impact of the Daily Mile program, since PE entails many other benefits that are probably not achieved by the Daily Mile (e.g. fundamental movement skill development, social benefits of sport, cognitive benefit of sport in terms of learning rules/strategy etc). Could elaborate this point slightly. 15. p 20 lines 17-18 – “Current evidence suggest that physical activity shows a positive effect on constructs related to academic achievement” – yes, but some studies are also finding that sedentary time is more strongly associated with academic achievement (e.g. Dumuid, D., Olds, T., Martín-Fernández, J. A., Lewis, L. K., Cassidy, L., & Maher, C. (2017). Academic performance and lifestyle behaviors in Australian school children: a cluster analysis. Health Education & Behavior, 44(6), 918-927. Maher, C., Lewis, L., Katzmarzyk, P. T., Dumuid, D., Cassidy, L., & Olds, T. (2016). The associations between physical activity, sedentary behaviour and academic performance. Journal of science and medicine in sport, 19(12), 1004-1009. – apologies for the self-citation – not suggesting you need to reference these). Could nuance this discussion point to acknowledge that we don’t understand the best balance between sedentary and PA for academic achievement. Some of the quotes from teachers in your study also alluded to this point.
--	---

REVIEWER	Sarah Kennedy
	University of Newcastle, School of Education
REVIEW RETURNED	27-Nov-2018

GENERAL COMMENTS	I thank the authors for the opportunity to review this manuscript. This is a well written manuscript, with the potential to make a contribution to the literature surrounding school-based physical activity interventions. However, in my belief the manuscript requires revisions prior to being accepted for publication. Firstly, the introduction does not set the scene for why this qualitative study was conducted, rather it is an introduction to the
---

	Daily Mile intervention and previous research. I implore the authors to conduct a search of literature in this space (see Nathan et al., 2017; Kennedy et al., 2018) to further strengthen the introduction and highlight the need for research in this space. Second, the discussion requires further support from the literature to justify points made. These points also need to be aligned with the introduction statements made, and link back to the importance of the study being presented - not the previous findings from the intervention as such. Finally, the manuscript requires a thorough grammatical check and language should be amended to suit an international audience.
--	---

VERSION 1 – AUTHOR RESPONSE

Reviewer 1.

This is an interesting paper, and reveals information which will be useful to inform future implementation of the Daily Mile program

We thank the reviewer for their helpful comments, our responses to each comment are detailed below:

1. Abstract – Structure of the abstract is a little unusual (there is no “methods” section – perhaps this is a journal requirement). At present, methodological detail is dispersed through other sections of the abstract (e.g. the study design is described in the objectives section, and the analytic approach is described in the results). Recommend moving these to create a methods section.

This abstract structure is a requirement of the journal unfortunately, we have tried to detail the methods throughout the abstract appropriately.

2. P3 line 4 - typo - teachers need apostrophe “teachers’

This change has now been made with an apostrophe added to teachers on P3 line 2.

3. St Ninian's primary school - capitalise primary school (LINE 24-5)

This change has now been made with primary school now capitalised P3 line 23

4 and 5. p 6 line 9-10 Primary school teachers who teach primaries one through seven at a school, which participated in the Daily Mile or had done so in the past were recruited from the participating schools. This sentence is rather awkwardly worded.

Thank you for the suggested rewording of the sentence, this has now been changed as suggested P6 lines 18-19.

6. Line 16 and other places – a teacher acted as a “gatekeeper” to others – this word is not appropriate for scientific writing. Suggest replacing throughout the manuscript with either liaison or facilitator.

Thank you for the suggested word change, we have now changed ‘gatekeeper’ to ‘liaison’ throughout the manuscript.

7. P 17 - "personally spoke to selected head-teachers" – on what basis were such head teachers selected? The methods should be reported in a level of detail that is reproducible.

We agree with the reviewer that this information was lacking in the manuscript. We have now added further clarification regarding the selection of head teachers to P7 lines 3-5.

8. P 7 – Data collection: How were the interviews recorded? What training did the interviewer have? Some more information about the interview procedure would be helpful

We agree with the reviewer that this information was lacking in the manuscript. We have now provided more information regarding how the interviews were recorded (with a portable audio recorder) P7 lines 21-22, and the expertise/level of experience of the interviewers (both experienced qualitative interviewers with formal training in evaluating public health interventions using qualitative methods; one to PhD level and both to masters level [MPH]) P7 lines 13-15.

9. P7 – data analysis: It appears that coding was determined based on the first two transcripts, and then applied to all remaining transcripts. I have never heard of this analysis approach. Please add citations to support this approach. How would this approach handle new topics that arise in subsequent transcripts that were not present in the first two transcripts?

Apologies. Upon re-reading the section in question it is clear why the reviewer thought that the coding framework was derived from only two transcripts and then applied to the rest. We have now added further clarification and re-worded this section to describe our approach to coding in adequate detail (P8 lines 2-13). To summarise, all transcripts were independently coded by both researchers. The initial coding framework was developed using the two longest transcripts, then as each remaining transcript was independently coded, the researchers discussed and agreed upon any further emerging themes. This has now been described in the manuscript. We have also added a reference regarding thematic analysis and code development to P8 line 2.

10. p 7 lines 16-17 "The analysis proceeded to the interpretive phase where similarities and differences between units of the data underwent dynamic series of negotiations before patterns of meaning and understanding developed" – this sentence is very jargony. What does this actually mean?

This sentence has now been deleted and replaced with the information added to P8 lines 2-13. Specifically "Interpretation of data was undertaken by comparing similarities and differences within and across themes".

11. P 6, line 13 - 11 women, 2 men. Abstract says 10 women.

Thank you for bringing this typo to our attention. We have now amended the abstract to correctly read 11 women.

12. P17, line 12 - schools should be lower case

This change has now been made to P17 line 21.

13. Discussion: paragraph 2 of the discussion is essentially the "implications" paragraph, and would make more sense if it was placed nearer the end of the discussion.

Thank you for this suggestion. We agree with the reviewer that this paragraph would make more sense placed later in the discussion. We have now moved it to come before the strengths/limitations section (P22 line 13- P23 line 3).

14. Paragraph 3 makes an interesting point about the Daily Mile apparently substituting PE for some teachers....

We thank the reviewer for this suggestion and we agree that this adds an interesting and important point to the argument, therefore we have incorporated this point into the discussion and added a reference (P20 line 11-14).

15. p 20 lines 17-18 – “Current evidence suggest that physical activity shows a positive effect on constructs related to academic achievement” – yes, but some studies are also finding that sedentary time is more strongly associated with academic achievement

We thank the reviewer for bringing this research to our attention, and we agree it is an important point to make in this paper. We have now added to the discussion to highlight the knowledge gap which currently exists regarding optimal PA and sedentary time balance with regards to academic achievement (P21 lines 13-19). We felt the references suggested by the reviewer illustrated this point well so have added them to the manuscript here.

Reviewer 2.

I thank the authors for the opportunity to review this manuscript. This is a well written manuscript, with the potential to make a contribution to the literature surrounding school-based physical activity interventions. However, in my belief the manuscript requires revisions prior to being accepted for publication....

We thank the reviewer for their helpful comments.
Our responses to their points are detailed below:

Firstly, the introduction does not set the scene for why this qualitative study was conducted, rather it is an introduction to the Daily Mile intervention and previous research

We agree the introduction could benefit from more emphasis on the need for this research, and we thank the reviewer for their suggested readings. We have now incorporated the findings of the systematic review by Nathan et al (2017) in to the introduction (P4 lines 14-18). Furthermore, we have also added further information on a recently published study of the factors influencing the Daily Mile's successful implementation (Ryde et al 2018). We have discussed the main findings and limitations of this study (P4 line 22 – P5 line 4), and have also referred to the author's recommendations for future research regarding the Daily Mile (P5 lines 4-6), specifically that future studies should 1. Target multiple geographic locations, 2. Interview more teachers who may not be as heavily invested in the daily mile. And 3. Target schools which had implemented the intervention with varying degrees of success. Given that our study has accounted for all three of these recommendations, we feel this addresses why this research was needed. We have now stated this in the manuscript (P5 lines 9-13).

Second, the discussion requires further support from the literature to justify points made. These points also need to be aligned with the introduction statements made, and link back to the importance of the study being presented - not the previous findings from the intervention as such.

Again we agree with the reviewer that the discussion would benefit from further reference to published work. We have mentioned how our study both supports and adds to the findings of Ryde et al (2018) with regards to our understanding of what influences successful implementation of the Daily Mile (P19

lines 1-5). As suggested, we have linked the findings of Nathan et al (2017) with our studies' findings with regards the lack of adequate facilities/adverse weather being a barrier to implementation (P22 lines 6-8). We have also referenced the suggested paper (Kennedy et al 2018) as it supports our findings that time constraints was a barrier to the implementation of a school-based PA initiative (P20 lines 24-25). We have also added further detail and references to our discussion points regarding PA vs. sedentary time and academic performance (P21 lines 13-19), and our points regarding the detrimental effects of combining PE time with the Daily Mile (P20 lines 11-14) which we feel give a more nuanced argument to these points.

Finally, the manuscript requires a thorough grammatical check and language should be amended to suit an international audience.

We have identified a number of grammatical errors which have been corrected throughout the manuscript. We have changed some phrases which are more relevant to UK to suit an international audience as suggested. (e.g. Converted mile in to kilometre (P3 lines 18-19); provided additional information on what age is served by primary school education (P3 lines 21-22).